# Ultra-Local Model Predictive Current Control of Permanent Magnet Synchronous Motor With Dual-Vector Based on Data-Driven Neural Networks

1st Chendong Zhao
*School of Marine Electrical Engineering*
*Dalian Maritime University*
Dalian, China
zhaochendong623@dlmu.edu.cn

2nd Dan Wang
*School of Marine Electrical Engineering*
*Dalian Maritime University*
Dalian, China
dwangdl@gmail.com

3rd Zhouhua Peng
*School of Marine Electrical Engineering*
*Dalian Maritime University*
Dalian, China
zhpeng@dlmu.edu.cn

4th Wenjie Wu
*School of Marine Electrical Engineering*
*Dalian Maritime University*
Dalian, China
wuwenjie@dlmu.edu.cn

*Abstract*—The traditional model predictive current control (MPCC) of permanent magnet synchronous motor (PMSM) has the advantages of simple control structure and excellent dynamic performance. However, the performance of the MPCC is significantly impacted by changes in motor parameters. The inside and outside unknown disturbance of the motor cause the parameters mismatch, which negatively affects the performance of the MPCC controller. To eliminate the effects of the parameters mismatch, the original parameter-based predictive model is replaced by an ultra-local model in this paper. An estimator based on data-driven neural network is designed to quickly and accurately estimate the total perturbation and control gain of the established ultra-local model. The proposed design solely utilizes the input and output information of the controlled system instead of relying on motor parameters, thus avoiding the negative effects of model parameters mismatch. In addition, the dual-vector mechanism and delay compensation are added to improve the control performance. Finally, the stability analysis is given, and simulated results show the availability of the proposed method.

*Index Terms*—permanent magnet synchronous motor (PMSM), data-driven neural network, dual-vector, parameters mismatch, ultra-local model

## I. INTRODUCTION

The permanent magnet synchronous motor (PMSM) is widely spreaded because of its superiority of high power density, compact size and excellent dynamic response [1]. However, the PMSM is a complex system with multiple variables, strong coupling, nonlinearity and variable parameters [2]. To achieve high performance and precision control, some advanced control methods, such as field-oriented control, direct torque control, and model predictive control (MPC) are used in the PMSM control system [3]. Among various control strategies, the MPC stands out in recent publications because of its simple structure and excellent control performance [4].

MPC uses the mathematical model of the control object to discretely obtain the predicted value and then optimizes the cost function to make the predicted value approach the expected value along the reference trajectory. MPC used in PMSM control field is classified into two types depending on the control variables: model predictive current control (MPCC) and model predictive torque control [5]. MPCC can improve dynamic performance by predicting the future current of the control system [6]. As a model-based method, the high control quality of the MPCC paradigm relies on the precise prediction of the controlled variable, i.e., the stator current of the PMSM. This inherited feature results in the model dependence issues of the MPC controller, i.e., the implementation of the MPCC relies on the accurate model parameters [7]. However, during motor operation, the parameters such as stator resistance, inductance, and magnetic flux are influenced by the ambient temperature and operating conditions, and the parameters mismatch increase the ripple of the torque and the stator current of the motor, which make the worse performance of the motor [8], [9]. Compared with the mismatch of the stator resistance and the magnetic flux, the MPCC is more significantly affected by the mismatch of the inductance [10].

To tackle the issue of model dependence in the MPCC method, many solutions are proposed. A Kalman Filter-based algorithm is proposed to learn motor parameters and decrease the current ripple due to parameters mismatch in [11]. Additionally, an online parameter identification strategy is proposed [12]. Furthermore, the observers are constructed to observe the disturbances of the current and compensate the model mismatch [13]. Despite the online identification strategies and observers can estimate parameters mismatch and provide feedback to the controller, those approaches exist large errors in dealing with severe mismatch of multiple parameters.

Apart from parameter identification-based methods, ultra-local model shows good potential to cope with the effects of model parameters variations and reduces parameter depen-

dence [14], [15]. So the ultra-local model is widely accepted to replace the original mathematical model of the motor to achieve high-performance control. In order to achieve accurate control, the total perturbation in the ultra-local model needs to be accurately estimated, thus various methods are developed [16]. A typical disturbance estimator is extended state observer (ESO), which is established to estimate system disturbances and exhibit superior performance [17]. However, the ultra-local model has an unavoidable dependence on control gain. The previous strategy has limitations that the impacts of the input voltage gain impacted by the parameters mismatch and the external perturbations is ignored and the gain is considered as a immutable constant [18].

To enhance the anti-interference capability and robustness of the ultra-local model based predictive current control, a novel dual-vector model predictive current control strategy based on ultra-local model by using data-driven neural network (DDNN-DVMPCC) is proposed in this paper. The proposed method uses the input and output information of the PMSM to establish an equivalent model based on ultra-local model, alleviating the dependence of MPCC method on accurate model parameters to address the issue of PMSM model mismatch arising from the parameters drift and system disturbance. A data-driven neural network-based disturbance estimator is established in this paper to estimate the total perturbation and the input voltage gain of the ultra-local model. This approach reduces the reliance on the parameters and mitigates the effect of parameters mismatch. Simultaneously, the proposed control strategy employs the mechanism of dual-vector and delay compensation to guarantee the precision of the current forecasts and enhance the current tracking performance. Simulation demonstrates that the proposed method obtains better control performance and parameters robustness than traditional control methods when dealing with parameters mismatch.

This article is structured as follows: The second section shows the mathematical model of PMSM. The third section shows the ultra-local model of the PMSM, the data-driven neural network based estimator and the dual-vector predictive current control with a delay compensation. In the fourth section, comparative simulations are conducted under different operating conditions with parameters mismatch. The simulated waveforms are provided and explained in detail. Finally, the conclusions are presented in the fifth section.

## II. MATHEMATICAL MODEL OF PMSM

In this research, the controlled motor is the surface permanent magnet synchronous motor (SPMSM) with the same equivalent inductance of the dq-axis, i.e., $L_d = L_q = L_s$. Consequently, the mathematical model of stator current in the rotating reference frame (dq-axis) can be written as

$$\begin{cases} u_d = R_s i_d + L_s \frac{di_d}{dt} - \omega_e L_s i_q \\ u_q = R_s i_q + L_s \frac{di_q}{dt} + \omega_e L_s i_d + \omega_e \psi_f \end{cases} \quad (1)$$

where $u_d$ and $u_q$ are the voltages of the dq-axis stator. $i_d$ and $i_q$ are the currents of the dq-axis stator. $L_s$ is the equivalent inductance of the dq-axis stator; $R_s$ is the equivalent resistance

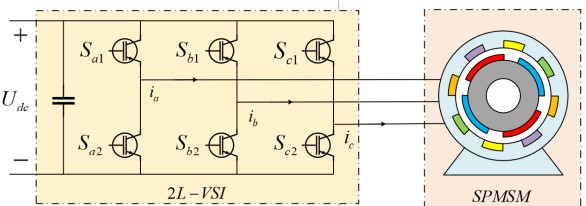

Fig. 1. Circuit digram of 2L-VSI and PMSM.

of the stator; $\omega_e$ is the electrical speed of the rotor; $\psi_f$ is the flux linkage.

In this research, the SPMSM is driven by a two-level voltage source inverter (2L-VSI), as illustrated in Fig. 1. The switching states generated by the 2L-VSI can be defined as

$$S = [S_a, S_b, S_c] \quad (2)$$

where $S = [000, 001, 010, 011, 100, 101, 110, 111]$.

Accordingly, the voltage can be calculated in accordance with the various switching states.

$$U_s = 2U_{dc}\left(S_a + S_b e^{j\frac{2\pi}{3}} + S_c e^{j\frac{4\pi}{3}}\right)/3 \quad (3)$$

where $U_{dc}$ is the voltage of the $DC$ bus.

## III. PROPOSED DDNN-DVMPCC ALGORITHM

In the conventional MPCC, the future current prediction is related to the motor physical parameters. During the operation of the motor, the motor parameters are influenced by the operating conditions and environmental factors, resulting in the parameters mismatch. The parameters mismatch increase the ripple of the torque and the stator current of the motor, which make the worse performance of the motor. Therefore, to enhance the anti-interference capability and robustness of the PMSM system against parameters mismatch, a control strategy for dual-vector model predictive current control based on ultra-local model by using data-driven neural network (DDNN-DVMPCC) is proposed.

### A. Ultra-Local Model

The first-order ultra-local model for the single-input and single-output system can be written as follows

$$\dot{x} = F + \alpha u \quad (4)$$

where $u$ and $x$ are the system variables of input and output, respectively; $F$ represents the total disturbance of the control system; $\alpha$ is the input gain.

Combined (4) with (1), the ultra-local model of SPMSM can be written as

$$\begin{cases} \frac{di_d}{dt} = F_d + \alpha u_d \\ \frac{di_q}{dt} = F_q + \alpha u_q \end{cases} \quad (5)$$

where $\alpha = \frac{1}{L_s}$, $F_q = -\frac{R_s}{L_s}i_q - \frac{1}{L_s}(L_s\omega_e i_d + \omega_e \psi_f)$, $F_d = -\frac{R_s}{L_s}i_d + \omega_e i_q$.

## B. Data-Driven Neural Network

To solve the problems of model uncertainty and unknown input gain, two estimators based on data-driven neural network are designed in this subsection.

The unknown functions $F_d$ and $F_q$ can be approximated by neural networks as follows:

$$\begin{cases} F_d = \Psi_d^T \sigma_d(\chi_d) + \epsilon_d \\ F_q = \Psi_q^T \sigma_q(\chi_q) + \epsilon_q \end{cases} \tag{6}$$

where $\chi_d$ and $\chi_q$ are inputs of neural networks. $\sigma_d(\chi_d)$ and $\sigma_q(\chi_q)$ are known activation functions; $\epsilon_d$ and $\epsilon_q$ are the approximation errors satisfying $\|\epsilon_d\| \le \epsilon_d^*$ and $\|\epsilon_q\| \le \epsilon_q^*$ with $\epsilon_d^*$ and $\epsilon_q^*$ being positive constants. $\Psi_d$ and $\Psi_q$ are the unknown weight matrices satisfying $\|\Psi_d\| \le \Psi_d^*$ and $\|\Psi_q\| \le \Psi_q^*$ with $\Psi_d^*$ and $\Psi_q^*$ being positive constants.

Let $\hat{i}_d$ and $\hat{i}_q$ be the estimation of $i_d$ and $i_q$; $\hat{\Psi}_d$ and $\hat{\Psi}_q$ be the estimation of $\Psi_d$ and $\Psi_q$; $\hat{\alpha}$ be the estimation of $\alpha$. Then, two data-driven neural network estimators are proposed as follows:

$$\begin{cases} \dot{\hat{i}}_d = \hat{\Psi}_d^T \sigma_d(\chi_d) + \hat{\alpha} u_d - k_d \tilde{i}_d \\ \dot{\hat{i}}_q = \hat{\Psi}_q^T \sigma_q(\chi_q) + \hat{\alpha} u_q - k_q \tilde{i}_q \end{cases} \tag{7}$$

where $\tilde{i}_d = (\hat{i}_d - i_d)$ and $\tilde{i}_q = (\hat{i}_q - i_q)$; $k_d$ and $k_q$ are positive constants.

According to the parallel learning method introduced in Ref. [19]. The update laws for $\hat{\Psi}_d$, $\hat{\Psi}_q$ and $\hat{\alpha}$ are designed by using historically accumulated data as follows:

$$\begin{bmatrix} \dot{\hat{\Psi}}_d \\ \dot{\hat{\alpha}} \end{bmatrix} = -\Gamma_d \text{Proj} \left\{ \begin{bmatrix} \hat{\Psi}_d \\ \hat{\alpha} \end{bmatrix}, \begin{bmatrix} \sigma_d(\chi_d) \\ u_d \end{bmatrix} \tilde{i}_d - k_{wd}\Phi_d \right\} \tag{8}$$

and

$$\begin{bmatrix} \dot{\hat{\Psi}}_q \\ \dot{\hat{\alpha}} \end{bmatrix} = -\Gamma_q \text{Proj} \left\{ \begin{bmatrix} \hat{\Psi}_q \\ \hat{\alpha} \end{bmatrix}, \begin{bmatrix} \sigma_q(\chi_q) \\ u_q \end{bmatrix} \tilde{i}_q - k_{wq}\Phi_q \right\} \tag{9}$$

where

$$\Phi_d = \sum_{k=1}^{k=p} \left\{ \begin{bmatrix} \sigma_d^k(\chi_d) \\ u_d^k \end{bmatrix} \left[ \dot{\hat{i}}_d - \hat{\Psi}_d^T \sigma_d^k(\chi_d) - \hat{\alpha} u_d^k \right] \right\} \tag{10}$$

and

$$\Phi_q = \sum_{k=1}^{k=p} \left\{ \begin{bmatrix} \sigma_q^k(\chi_q) \\ u_q^k \end{bmatrix} \left[ \dot{\hat{i}}_q - \hat{\Psi}_q^T \sigma_q^k(\chi_q) - \hat{\alpha} u_q^k \right] \right\} \tag{11}$$

where Proj denotes the projection operator [20]; $\Gamma_d$, $\Gamma_q$ and $k_{wd}$, $k_{wq}$ are positive constants; $p$ is a positive integer number that denotes the length of the memory stack; $\sigma_d^k(\chi_d)$, $\sigma_q^k(\chi_q)$ and $u_d^k$, $u_q^k$ are the historical information stored at each $k$ instant.

Next, the stability of the proposed data-driven neural network is analyzed. The estimation errors are defined as

$$\begin{cases} \tilde{i}_d = \hat{i}_d - i_d, \tilde{i}_q = \hat{i}_q - i_q \\ \tilde{\Psi}_d = \hat{\Psi}_d - \Psi_d, \tilde{\Psi}_q = \hat{\Psi}_q - \Psi_q \\ \tilde{\alpha} = \hat{\alpha} - \alpha \end{cases} \tag{12}$$

The error dynamic equation of $\tilde{i}_d$ and $\tilde{i}_q$ can be expressed as follows

$$\begin{cases} \dot{\tilde{i}}_d = \tilde{\Psi}_d^T \sigma_d(\chi_d) + \tilde{\alpha} u_d - k_d \tilde{i}_d + \epsilon_d \\ \dot{\tilde{i}}_q = \tilde{\Psi}_q^T \sigma_q(\chi_q) + \tilde{\alpha} u_q - k_q \tilde{i}_q + \epsilon_q \end{cases} \tag{13}$$

In addition, the error dynamics of $\tilde{\Psi}_d$, $\tilde{\Psi}_q$ and $\tilde{\alpha}$ are given by

$$\begin{bmatrix} \dot{\tilde{\Psi}}_d \\ \dot{\tilde{\alpha}} \end{bmatrix} = -\Gamma_d \text{Proj} \left\{ \begin{bmatrix} \tilde{\Psi}_d \\ \tilde{\alpha} \end{bmatrix}, \begin{bmatrix} \sigma_d(\chi_d) \\ u_d \end{bmatrix} \tilde{i}_d - k_{wd}\Phi_d \right\} \tag{14}$$

and

$$\begin{bmatrix} \dot{\tilde{\Psi}}_q \\ \dot{\tilde{\alpha}} \end{bmatrix} = -\Gamma_q \text{Proj} \left\{ \begin{bmatrix} \tilde{\Psi}_q \\ \tilde{\alpha} \end{bmatrix}, \begin{bmatrix} \sigma_q(\chi_q) \\ u_q \end{bmatrix} \tilde{i}_q - k_{wq}\Phi_q \right\} \tag{15}$$

The subsystem (13)–(15), are viewed as a system with the states being $\tilde{i}_d$, $\tilde{i}_q$, $\tilde{\Psi}_d$, $\tilde{\Psi}_q$ and $\tilde{\alpha}$, the inputs being $\epsilon_d$ and $\epsilon_q$ is input-to-state stable.

Proof: Construct a Lyapunov function candidate as

$$\begin{aligned} V = & \frac{1}{2}\tilde{i}_d^2 + \frac{1}{2}\tilde{i}_q^2 + \frac{1}{2} \begin{bmatrix} \tilde{\Psi}_d \\ \tilde{\alpha} \end{bmatrix}^T \Gamma_d^{-1} \begin{bmatrix} \tilde{\Psi}_d \\ \tilde{\alpha} \end{bmatrix} \\ & + \frac{1}{2} \begin{bmatrix} \tilde{\Psi}_q \\ \tilde{\alpha} \end{bmatrix}^T \Gamma_q^{-1} \begin{bmatrix} \tilde{\Psi}_q \\ \tilde{\alpha} \end{bmatrix} \end{aligned} \tag{16}$$

Then, the derivative of the Lyapunov function (16) can be written as

$$\begin{aligned} \dot{V} = & \tilde{i}_d \dot{\tilde{i}}_d + \tilde{i}_q \dot{\tilde{i}}_q + \begin{bmatrix} \tilde{\Psi}_d \\ \tilde{\alpha} \end{bmatrix}^T \Gamma_d^{-1} \begin{bmatrix} \dot{\tilde{\Psi}}_d \\ \dot{\tilde{\alpha}} \end{bmatrix} \\ & + \begin{bmatrix} \tilde{\Psi}_q \\ \tilde{\alpha} \end{bmatrix}^T \Gamma_q^{-1} \begin{bmatrix} \dot{\tilde{\Psi}}_q \\ \dot{\tilde{\alpha}} \end{bmatrix} \\ = & \tilde{i}_d(\tilde{\Psi}_d^T \sigma_d(\chi_d) + \tilde{\alpha} u_d - k_d \tilde{i}_d + \epsilon_d) + \tilde{i}_q(\tilde{\Psi}_q^T \sigma_q(\chi_q) \\ & + \tilde{\alpha} u_q - k_q \tilde{i}_q + \epsilon_q) \\ & - \begin{bmatrix} \tilde{\Psi}_d \\ \tilde{\alpha} \end{bmatrix}^T \Gamma_d^{-1} \Gamma_d \left\{ \begin{bmatrix} \sigma_d(\chi_d) \\ u_d \end{bmatrix} \tilde{i}_d - k_{wd}\Phi_d \right\} \\ & - \begin{bmatrix} \tilde{\Psi}_q \\ \tilde{\alpha} \end{bmatrix}^T \Gamma_q^{-1} \Gamma_q \left\{ \begin{bmatrix} \sigma_q(\chi_q) \\ u_q \end{bmatrix} \tilde{i}_q - k_{wq}\Phi_q \right\} \\ = & -k_d \tilde{i}_d^2 + \tilde{i}_d \epsilon_d - k_q \tilde{i}_q^2 + \tilde{i}_q \epsilon_q \\ & - k_{wd} \begin{bmatrix} \tilde{\Psi}_d \\ \tilde{\alpha} \end{bmatrix}^T \Phi_d - k_{wq} \begin{bmatrix} \tilde{\Psi}_q \\ \tilde{\alpha} \end{bmatrix}^T \Phi_q \\ = & -k_d \tilde{i}_d^2 + \tilde{i}_d \epsilon_d - k_q \tilde{i}_q^2 + \tilde{i}_q \epsilon_q \\ & - k_{wd} \begin{bmatrix} \tilde{\Psi}_d \\ \tilde{\alpha} \end{bmatrix}^T \\ & \sum_{k=1}^{k=p} \left\{ \begin{bmatrix} \sigma_d^k(\chi_d) \\ u_d^k \end{bmatrix} \left[ \dot{\hat{i}}_d - \hat{\Psi}_d^T \sigma_d^k(\chi_d) - \hat{\alpha} u_d^k \right] \right\} \\ & - k_{wq} \begin{bmatrix} \tilde{\Psi}_q \\ \tilde{\alpha} \end{bmatrix}^T \\ & \sum_{k=1}^{k=p} \left\{ \begin{bmatrix} \sigma_q^k(\chi_q) \\ u_q^k \end{bmatrix} \left[ \dot{\hat{i}}_q - \hat{\Psi}_q^T \sigma_q^k(\chi_q) - \hat{\alpha} u_q^k \right] \right\} \end{aligned} \tag{17}$$

Since the activation function of the neural network and the control input are bounded, there exists $\sigma_d^*(\chi_d)$, $\sigma_q^*(\chi_q)$, $u_d^*$ and $u_q^*$ satisfying that $\|\sigma_d(\chi_d)\| \leq \sigma_d^*(\chi_d)$, $\|\sigma_q(\chi_q)\| \leq \sigma_q^*(\chi_q)$, $\|u_d\| \leq u_d^*$ and $\|u_q\| \leq u_q^*$.

Then it can be obtained that

$$
\begin{aligned}
\dot{V} \leq & -k_d \left\|\tilde{i}_d\right\|^2 + \left\|\tilde{i}_d\right\| \|\epsilon_d\| - k_q \left\|\tilde{i}_q\right\|^2 + \left\|\tilde{i}_q\right\| \|\epsilon_q\| \\
& - pk_{wd}\sigma_d^{*2}(\chi_d) \left\|\tilde{\Psi}_d\right\|^2 - pk_{wd}u_d^{*2} \|\tilde{\alpha}\|^2 \\
& + pk_{wd}\sigma_d^*(\chi_d) \left\|\tilde{\Psi}_d\right\| \|\epsilon_d\| + pk_{wd}u_d^* \|\tilde{\alpha}\| \|\epsilon_d\| \\
& - pk_{wq}\sigma_q^{*2}(\chi_q) \left\|\tilde{\Psi}_q\right\|^2 - pk_{wq}u_q^{*2} \|\tilde{\alpha}\|^2 \\
& + pk_{wq}\sigma_q^*(\chi_q) \left\|\tilde{\Psi}_q\right\| \|\epsilon_q\| + pk_{wq}u_q^* \|\tilde{\alpha}\| \|\epsilon_q\| \\
\leq & -h_1 \|S\|^2 + h_2 \|S\| \|M\|
\end{aligned}
\tag{18}
$$

where $h_1 = min(k_d, k_q, pk_{wd}\sigma_d^{*2}(\chi_d), pk_{wq}\sigma_q^{*2}(\chi_q), pk_{wd}u_d^{*2}, pk_{wd}u_q^{*2})$, $h_2 = max(1, pk_{wd}\sigma_d^*(\chi_d), pk_{wq}\sigma_q^*(\chi_q), pk_{wd}u_d^*, pk_{wq}u_q^*)$, $S = [\|\tilde{i}_d\|, \|\tilde{i}_q\|, \|\tilde{\Psi}_d\|, \|\tilde{\Psi}_q\|, \|\tilde{\alpha}\|]^T$, $M = [\|\epsilon_d\|, \|\epsilon_q\|]^T$.

If $\|S\|$ satisfies the following inequality

$$
\|S\| \geq \frac{h_2 \|M\|}{2h_1}
\tag{19}
$$

Then the above inequality (18) can be further simplified into the following form

$$
\dot{V} \leq -\frac{1}{2}h_1 \|S\|^2
\tag{20}
$$

It can be concluded that the error system is input-to-state stable. In addition, there exists an $\mathcal{KL}$ function $\varphi(\cdot)$ and $\mathcal{K}_\infty$ functions $\phi^{\epsilon_d}(\cdot)$ and $\phi^{\epsilon_q}(\cdot)$, such that $\|S(t)\|$ satisfies the following inequality

$$
\|S(t)\| \leq \varphi(\|S(t_0)\|, t - t_0) + \phi^{\epsilon_d}(\|\epsilon_d\|) + \phi^{\epsilon_q}(\|\epsilon_q\|)
\tag{21}
$$

where the specific forms of $\phi^{\epsilon_d}(\cdot)$ and $\phi^{\epsilon_q}(\cdot)$ are as follows

$$
\phi^{\epsilon_d}(s) = \phi^{\epsilon_q}(s) = \frac{sh_2 \sqrt{\lambda_{max}(K)}}{2h_1 \sqrt{\lambda_{min}(K)}}
\tag{22}
$$

where $K = diag\left\{1, \Gamma_d^{-1}, \Gamma_q^{-1}\right\}$.

From the above design process, it can be seen that the data-driven neural network estimators designed in this chapter can simultaneously obtain the model uncertainty and the unknown control gain.

### C. Dual-Vector MPCC Controller

Fig. 2 illustrates the structure of the PMSM system controller which consists of two main components: ultra-local model based predictive current control by using data-driven neural network and dual-vector mechanism.

In this part, we use the data-driven neural network estimator from the previous section to estimate all the necessary information including the uncertainty of the motor system and the unknown input gain. And the estimated variables are sent to the current controller. The predictive current controller is

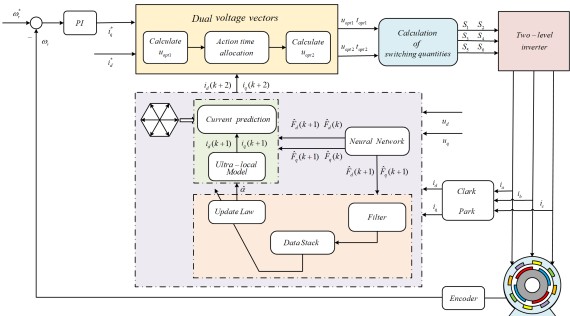

Fig. 2. The structure of DDNN-DVMPCC.

designed based on the principle of dual vector with a delay compensation.

The first-order Euler discrete method is used to obtain the discrete ultra-local model:

$$
\begin{cases}
\hat{i}_d(k+1) = i_d(k) + T_s[\hat{F}_d(k) + \hat{\alpha}u_d(k)] \\
\hat{i}_q(k+1) = i_q(k) + T_s[\hat{F}_q(k) + \hat{\alpha}u_q(k)]
\end{cases}
\tag{23}
$$

where $\hat{F}_d(k)$ and $\hat{F}_q(k)$ are the observations of the $(k)$ moment. $u_d(k)$ and $u_q(k)$ are the input of the $(k)$ moment.

Actually the voltage vector under single-step forecasting may not be optimal and even make the performance worse due to the existence of the delay. Consequently, using two-step forecasting with a delay compensation is necessary to controller. The predicted dq-axis currents with a delay compensation can be written as

$$
\begin{cases}
\hat{i}_d(k+2) = \hat{i}_d(k+1) + T_s[\hat{F}_d(k+1) \\
\qquad\qquad + \hat{\alpha}u_d(k+1)] \\
\hat{i}_q(k+2) = \hat{i}_q(k+1) + T_s[\hat{F}_q(k+1) \\
\qquad\qquad + \hat{\alpha}u_q(k+1)]
\end{cases}
\tag{24}
$$

The principle of selecting the switching state is to traverse the corresponding voltage vectors in all switching states. Subsequently, the switching state corresponding to the predicted value closest to the reference is selected according to the principle of minimum difference. The cost function is as follows

$$
Cf_j = [i_d^{ref} - \hat{i}_d(k+2) \mid_j]^2 + [i_q^{ref} - \hat{i}_q(k+2) \mid_j]^2
\tag{25}
$$

where the d-axis reference current $i_d^{ref}$ is configured to zero; the q-axis reference current $i_q^{ref}$ is configured by the speed controller of the outside loop.

The traditional model predictive control make performance worse due to the single-vector is acted in the entire control cycle. Therefore, the dual-vector modulation strategy is added in the proposed DDNN-DVMPCC method for decreasing the current ripple. Two-vector synthesis as illustrated in Fig. 3.

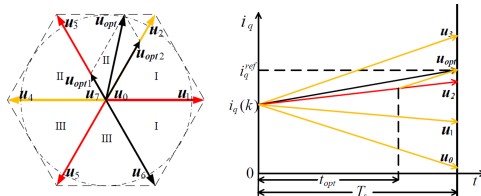

Fig. 3. Selection and synthesis of voltage vector.

Two optimal voltage vectors are calculated in one sampling period. According to the synthesis of the vectors, the resultant vector $u_{d\_opt}$ and $u_{q\_opt}$ can be expressed as

$$\begin{cases} u_{d\_opt}(k+1) = \dfrac{t_{opt1}}{T_s} u_{d\_opt1}(k+1) \\ \qquad\qquad + \dfrac{(T_s - t_{opt1})}{T_s} u_{dj}(k+1) \\ u_{q\_opt}(k+1) = \dfrac{t_{opt1}}{T_s} u_{q\_opt1}(k+1) \\ \qquad\qquad + \dfrac{(T_s - t_{opt1})}{T_s} u_{qj}(k+1) \end{cases} \quad (26)$$

where $t_{opt1}$ is the action time for the first optimal voltage vector $u_{opt1}$; $u_{d\_opt1}$ and $u_{q\_opt1}$ are the dq-axis voltage components for $u_{opt1}$. $u_{dj}$ and $u_{qj}$ are the dq-axis voltage components for $u_j$, where $(j = 0, 1, 2, ..., 7)$.

$u_{opt1}$ is selected based on the traditional MPCC among eight voltage vectors. To calculate the second optimal voltage vector $u_{opt2}$, $u_{opt1}$ is combined with each of the eight basic voltage vectors.

Substituting the synthesised voltage vector into (24), the predicted current at the (k+2) moment when acting on the synthesised voltage vector is

$$\begin{cases} \hat{i}_{d\_opt}(k+2) = \hat{i}_d(k+1) + T_s[\hat{F}_d(k+1) \\ \qquad\qquad + \hat{\alpha} u_{d\_opt}(k+1)] \\ \hat{i}_{q\_opt}(k+2) = \hat{i}_q(k+1) + T_s[\hat{F}_q(k+1) \\ \qquad\qquad + \hat{\alpha} u_{q\_opt}(k+1)] \end{cases} \quad (27)$$

The optimal voltage vector combination is selected by using the cost function shown in (25).

$$Cf_j = [i_d^{ref} - \hat{i}_{d\_opt}(k+2)\mid_j]^2 + [i_q^{ref} - \hat{i}_{q\_opt}(k+2)\mid_j]^2 \quad (28)$$

Therefore, the second optimal voltage vector $u_{opt2}$ is the $u_j$ in the combination of voltage vectors corresponding to the current value that minimises the cost function.

In this research, using the q-axis current without differential beats to obtain the switching time of $u_{opt1}$ and $u_j$ in the sampling period, which must be satisfied

$$i_q(k+1) = i_q(k) + k_{opt1} t_{opt1} + k_j(T_s - t_{opt1}) = i_q^* \quad (29)$$

According to (18), the operating time of $u_{opt1}$ can be expressed as

$$t_{opt1} = \frac{(i_q^* - i_q(k) - k_j T_s)}{k_{opt1} - k_j} \quad (30)$$

where $k_{opt1}$ and $k_j$ are the slopes of $i_q$ under $u_{opt1}$ and $u_j$. The duration of action of $u_j$ is $(T_s - t_{opt1})$. $k_{opt1}$ and $k_j$ can be expressed as

$$k_{opt1} = \frac{di_q}{dt}\mid_{u_q = u_{q\_opt1}} = \hat{F}_q + \hat{\alpha} u_{q\_opt1} \quad (31)$$

$$k_j = \frac{di_q}{dt}\mid_{u_q = u_{qj}} = \hat{F}_q + \hat{\alpha} u_{qj} \quad (32)$$

If the difference between the adjacent voltage vectors is small, $t_{opt1}$ may not be within the sampling period $T_s$, Therefore, it is necessary to set the limit of the optimal duration $t_{opt1}$ to ensure the prediction of the future current.

$$\begin{cases} 0, t_{opt1} < 0 \\ t_{opt1}, 0 \leq t_{opt1} \leq T_s \\ T_s, t_{opt1} > T_s \end{cases} \quad (33)$$

Through the dual-vector modulation algorithm, the cost function selects the switching state corresponding to the predicted current value closest to the reference value and the optimal vector switching time to synthesise the optimal voltage vector. This is equivalent to taking into account the impact of the operating time in the cost function. Not only the voltage vector is optimal, but also the operating time is optimal.

## IV. SIMULATION VERIFICATION RESULTS

This part establishes a simulation model in order to verify the effectiveness of the proposed algorithm through the comparative simulation experiments. The SPMSM model parameters are given in Table 1.

To demonstrate the superiority of the proposed strategy, a series of comparative simulations are built. Under the condition of motor parameters mismatch, the simulation under the speed step and load step at 1000 r/min is constructed. The three-phase currents and torque are compared. Furthermore, the tracking waveform of the total disturbance and the control gain of the system model estimated by the data-driven neural network predictor are evaluated.

The impact of various factors and unknown disturbances during motor operation has resulted in the motor parameters and the controller parameters mismatch, which makes the control performance worse. Because of changes in inductance parameters cause a greater effect on control performance than changes in resistance and flux. Consequently, the simulation conditions are created to simulate and compare the two control strategies under the condition of 0.5 $L_s$.

The current and torque waveforms of the two strategies under different operational conditions are indicated below.

TABLE I
SPMSM PARAMETERS

| Parameter | Symbol | Value |
|---|---|---|
| DC voltage $(V)$ | $U_{dc}$ | **311** |
| Number of pole pairs | $p$ | **4** |
| Stator resistance $(\Omega)$ | $R_s$ | **0.958** |
| Stator inductance $(H)$ | $L_s$ | **0.00525** |
| Flux linkage $(Wb)$ | $\psi_f$ | **0.1827** |

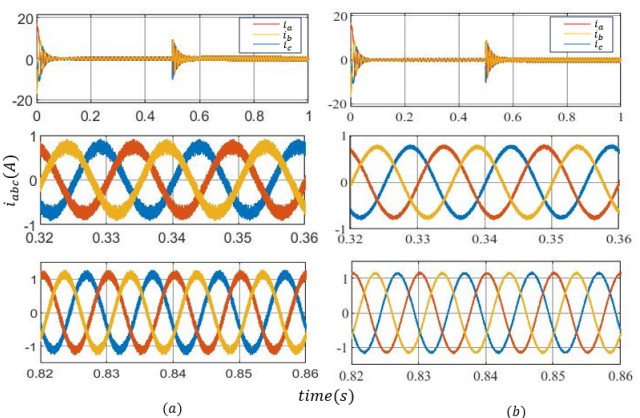

Fig. 4. Three-phase current with $50\%L_s$ under speed step. (a) FCS-MPCC. (b) Proposed DDNN-DVMPCC.

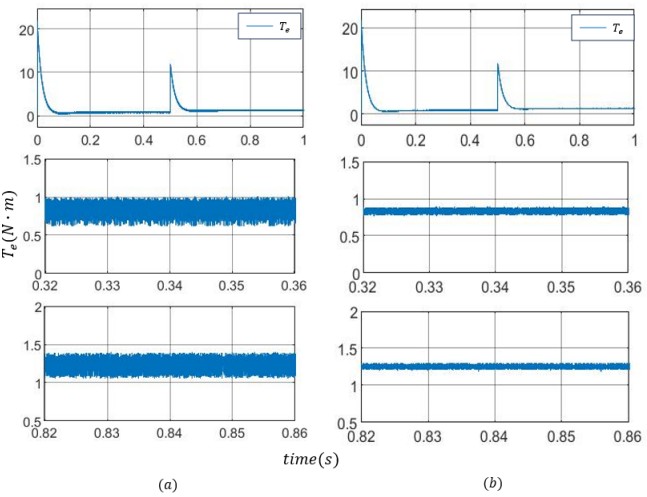

Fig. 5. $T_e$ with $50\%L_s$ under speed step. (a) FCS-MPCC. (b) Proposed DDNN-DVMPCC.

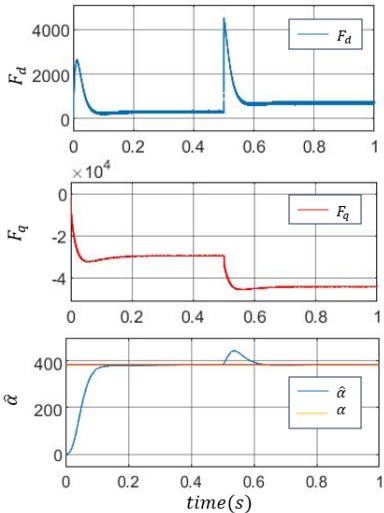

Fig. 6. Observed values of $\alpha$, $F_d$ and $F_q$ with $50\%L_s$ under speed step.

Fig. 4 shows the three-phase current simulation waveforms of the two control strategies under speed step 1000 r/min to 1500 r/min with $50\%L_s$. According to the waveform, the proposed DDNN-DVMPCC control strategy is smaller than that of MPCC in terms of current ripple.

Fig. 5 shows the torque simulation waveforms for the two control strategies under speed step 1000 r/min to 1500 r/min with $50\%L_s$. According to the waveform, the proposed DDNN-DVMPCC control strategy is better than that of MPCC in terms of torque ripple.

Fig. 6 shows the output values of the data-driven neural network estimator under speed step 1000 r/min to 1500 r/min with $50\%L_s$. According to the waveform, the proposed DDNN-DVMPCC control strategy can accurately estimate the unknown function and control gain under parameter perturbation.And the estimated value is quickly adjusted and stabilised to the exact value when the motor operating status changes.

Fig. 7 shows the three-phase current simulation waveforms of the two control strategies under load step 0 N.m to 5 N.m with $50\%L_s$. According to the waveform, the proposed DDNN-DVMPCC control strategy is smaller than that of MPCC in terms of current ripple.

Fig. 8 shows the torque simulation waveforms for the two control strategies under load step 0 N.m to 5 N.m with $50\%L_s$. According to the waveform, the proposed DDNN-DVMPCC control strategy is better than that of MPCC in terms of torque ripple.

Fig. 9 shows the output values of data-driven neural network estimator under load step 0 N.m to 5 N.m with $50\%L_s$. It indicated that the proposed DDNN-DVMPCC can accurately estimate the unknown function and control gain under parameter perturbation. And the estimated value is quickly adjusted and stabilised to the exact value when the motor operating status changes.

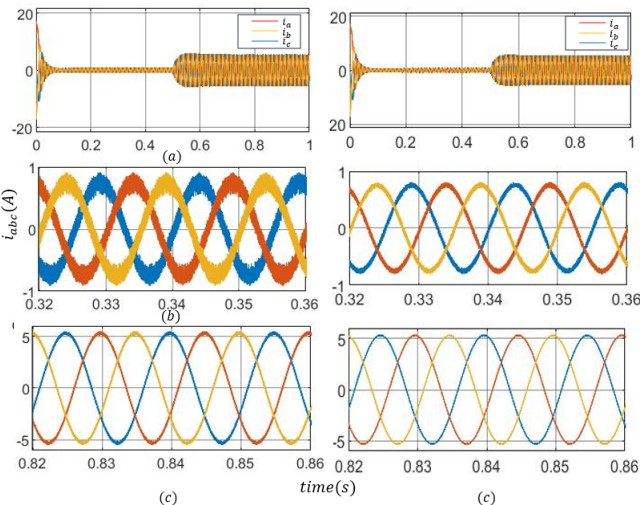

Fig. 7. Three-phase current with $50\%L_s$ under load step. (a) FCS-MPCC. (b) Proposed DDNN-DVMPCC.

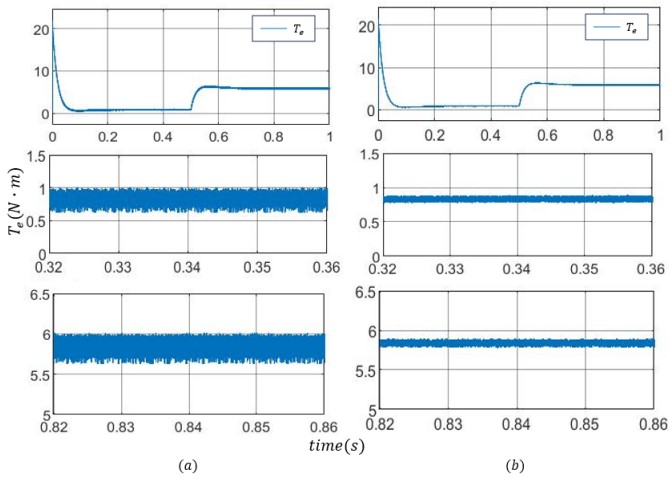

Fig. 8. $T_e$ with $50\%L_s$ under load step. (a) FCS-MPCC. (b) Proposed DDNN-DVMPCC.

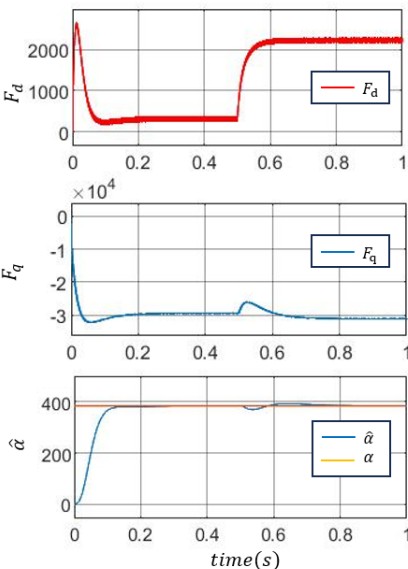

Fig. 9. Observed values of $\alpha$, $F_d$ and $F_q$ with $50\%L_s$ under Load step.

## ACKNOWLEDGMENT

In this paper, an ultra-local model based predictive current control with dual-vector based on data-driven neural network for permanent magnet synchronous motor is studied. This method eliminates the dependence of conventional MPC method has on the model parameters. Furthermore, the addition of dual-vector mechanism and delay compensation enable the system to obtain excellent steady-state performance. Under parameters mismatch, the proposed control strategy can quickly and accurately estimate the unknown function and control gain, and feed back to the controller. The simulation illustrates that the proposed control strategy obtains excellent current tracking performance, reduces the ripple of current and torque, and enhances parameter robustness than MPCC.

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
