# OpenReview forum: "Ultra-Local Model Predictive Current Control of Permanent Magnet Synchronous Motor With Dual-Vector Based on Data-Driven Neural Networks"
_IEEE.org/ICIST/2024/Conference — IEEE ICIST 2024 Conference Submission_

### Official Review · Reviewer_bdhm · 2024-08-21
**accept**

**Rating:** 7
**Confidence:** 3

**Review:**

Comment: This paper studies an ultra-local model based predictive current control with dual-vector based on data-driven neural network for permanent magnet synchronous motor. The theory is correct and can be accepted after responding the following comments.
(1) More comprehensive literature review is needed to clarify the research gap and research motivation.
(2) In(22), the author does not explain λ in it.
(3) In the end of the conclusions, some research directions are suggested to be added.

---

### Official Review · Reviewer_oYPC · 2024-08-23
**this work is well organized and appears potentially interesting, it can be accepted with a little modification.**

**Rating:** 7
**Confidence:** 3

**Review:**

The paper makes significant strides in addressing the challenges associated with parameter mismatch in MPCC for PMSMs. By employing an ultra-local model and leveraging data-driven neural networks, the authors successfully reduce the dependence on accurate motor parameters, a critical issue in high-performance motor control. The inclusion of dual-vector control and delay compensation further strengthens the proposed approach, making it a robust solution with better dynamic performance compared to traditional methods. The use of simulations to validate the method adds credibility, though experimental validation would further solidify the findings. Overall, the work represents a valuable contribution to the field of motor control, particularly in applications requiring high precision and robustness against parameter variations. In general, this work is well organized and appears potentially interesting, it can be accepted with a little modification.
1.	Improper use of articles or long sentence structures with separators might make it difficult to follow the paper. For example, the use of ‘the’ in the paper and the presentation of subordinate clauses. Please check the full text again and modify the grammar problems.
2.	Please highlight the contributions of the paper.
3.	Please add the necessary comments for Figures.
4.	Meanwhile, please elaborate on the future plans.

---

### Official Review · Reviewer_ZEfA · 2024-08-26
**Ultra-Local Model Predictive Current Control of Permanent Magnet Synchronous Motor With Dual-Vector Based on Data-Driven Neural Networks**

**Rating:** 7
**Confidence:** 2

**Review:**

This paper investigated the traditional model predictive current control (MPCC) of permanent magnet synchronous motor (PMSM). The obtained result is valuable and can be accepted if the following problems can be clarified.
1. The paper should include comparisons against the existing literature to demonstrate its advantages.
2. The paper should be added to the Assumptions and definitions with relative references to show the rationality of this paper.
3. This paper uses the algorithm of neural network. What are the advantages of neural network?

---

### Decision · Program_Chairs · 2024-09-06

Accept (Oral)